# WORLD MODEL AS A GRAPH:
# LEARNING LATENT LANDMARKS FOR PLANNING

## ABSTRACT

Planning, the ability to analyze the structure of a problem in the large and decompose it into interrelated subproblems, is a hallmark of human intelligence. While deep reinforcement learning (RL) has shown great promise for solving relatively straightforward control tasks, it remains an open problem how to best incorporate planning into existing deep RL paradigms to handle increasingly complex environments. One prominent framework, Model-Based RL, learns a world model and plans using step-by-step virtual rollouts. This type of world model quickly diverges from reality when the planning horizon increases, thus struggling at long-horizon planning. How can we learn world models that endow agents with the ability to do temporally extended reasoning? In this work, we propose to learn graph-structured world models composed of sparse, multi-step transitions. We devise a novel algorithm to learn latent landmarks that are scattered (in terms of reachability) across the goal space as the nodes on the graph. In this same graph, the edges are the reachability estimates distilled from Q-functions. On a variety of high-dimensional continuous control tasks ranging from robotic manipulation to navigation, we demonstrate that our method, named $L^3P$, significantly outperforms prior work, and is oftentimes the only method capable of leveraging both the robustness of model-free RL and generalization of graph-search algorithms. We believe our work is an important step towards scalable planning in reinforcement learning.

## 1 INTRODUCTION

An intelligent agent should be able to solve difficult problems by breaking them down into sequences of simpler problems. Classically, planning algorithms have been the tool of choice for endowing AI agents with the ability to reason over complex long-horizon problems (Doran & Michie, 1966; Hart et al., 1968). Recent years have seen an uptick in monographs examining the intersection of classical planning techniques – which excel at temporal abstraction – with deep reinforcement learning (RL) algorithms – which excel at state abstraction. Perhaps the ripest fruit born of this relationship is the AlphaGo algorithm, wherein a model free policy is combined with a MCTS (Coulom, 2006) planning algorithm to achieve superhuman performance on the game of Go (Silver et al., 2016a).

In the field of robotics, progress on combining planning and reinforcement learning has been somewhat less rapid, although still resolute. Indeed, the laws of physics in the real world are infinitely more complex than the simple rules of Go. Unlike board games such as chess and Go, which have deterministic and known dynamics and discrete action space, robots have to deal with a probabilistic and unpredictable world, and the action space for robots is oftentimes continuous. As a result, planning in robotics presents a much harder problem. One general class of methods (Sutton, 1991) seeks to combine model-based planning and deep RL. These methods can be thought of as an extension of model-predictive control (MPC) algorithms, with the key difference being that the agent is trained over hypothetical experience in addition to the actually collected experience. The primary shortcoming of this class of methods is that, like MCTS in AlphaGo, they resort to planning with action sequences – forcing the robot to plan for each action at every hundred milliseconds. Planning on the level of action sequences is fundamentally bottlenecked by the accuracy of the learned dynamics model and the horizon of a task, as the learned world model quickly diverges over a long horizon. This limitation shows that world models in the traditional Model-based RL (MBRL) setting often fail to deliver the promise of planning.

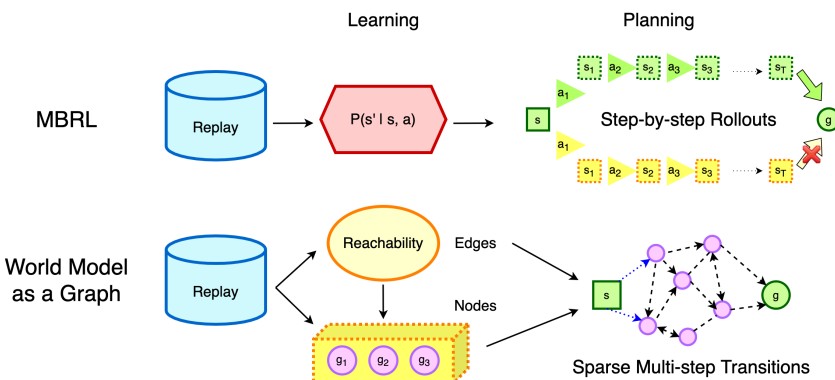

Figure 1: MBRL versus $L^3P$ (World Model as a Graph). MBRL does step-by-step virtual rollouts with the world model and quickly diverges from reality when the planning horizon increases. $L^3P$ models the world as a graph of sparse multi-step transitions, where the nodes are learned latent landmarks and the edges are reachability estimates. $L^3P$ succeeds at temporally extended reasoning.

Another general class of methods, Hierarchical RL (HRL), introduces a higher-level learner to address the problem of planning (Dayan & Hinton, 1993; Vezhnevets et al., 2017; Nachum et al., 2018). In this scenario, a goal-based RL agent serves as the worker, and a manager learns what sequences of goals it must set for the worker to achieve a complex task. While this is apparently a sound solution to the problem of planning, hierarchical learners neither explicitly learn a higher-level model of the world nor take advantage of the graph structure inherent to the problem of search.

To better combine classical planning and reinforcement learning, we propose to learn graph-structured world models composed of sparse multi-step transitions. To model the world as a graph, we borrow a concept from the navigation literature – the idea of landmarks (Wang et al., 2008). Landmarks are essentially states that an agent can navigate between in order to complete tasks. However, rather than simply using previously seen states as landmarks, as is traditionally done, we will instead develop a novel algorithm to learn the landmarks used for planning. Our key insight is that by mapping previously achieved goals into a latent space that captures the temporal distance between goals, we can perform clustering in the latent space to group together goals that are easily reachable from one another. Subsequently, we can then decode the latent centroids to obtain a set of goals scattered (in terms of reachability) across the goal space. Since our learned landmarks are obtained from latent clustering, we call them *latent landmarks*. The chief algorithmic contribution of this paper is a new method for planning over learned latent landmarks for high-dimensional continuous control domains, which we name Learning Latent Landmarks for Planning ($L^3P$).

The idea of reducing planning in RL to a graph search problem has enjoyed some attention recently (Savinov et al., 2018a; Eysenbach et al., 2019; Huang et al., 2019; Liu et al., 2019; Yang et al., 2020; Laskin et al., 2020). A key difference between those works and $L^3P$ is that our use of latent landmarks allows us to substantially reduce the size of the search space. What's more, we make improvements to the graph search module and the online planning algorithm to improve the robustness and sample efficiency of our method. As a result of those decisions, our algorithm is able to achieve superior performance on a variety of robotics domains involving both navigation and manipulation. In addition to the results presented in Section 5, videos of our algorithm's performance, and an analysis of the sub-tasks discovered by the latent landmarks, may be found at `https://sites.google.com/view/latent-landmarks/`.

## 2 RELATED WORKS

The problem of learning landmarks to aid in robotics problems has a long and rich history (Gillner & Mallot, 1998; Wang & Spelke, 2002; Wang et al., 2008). Prior art has been deeply rooted in the classical planning literature. For example, traditional methods would utilize Dijkstra et al. (1959) to plan over generated waypoints, SLAM (Durrant-Whyte & Bailey, 2006) to simultaneously integrate mapping, or the RRT algorithm (LaValle, 1998) for explicit path planning. The A* algorithm (Hart

et al., 1968) further improved the computational efficiency of Dijkstra. Those types of methods often heavily rely on a hand-crafted configuration space that provides prior knowledge.

Planning is intimately related to model-based RL (MBRL), as the core ideas underlying learned models and planners can enjoy considerable overlap. Perhaps the most clear instance of this overlap is Model Predictive Control (MPC), and the related Dyna algorithm (Sutton, 1991). When combined with modern techniques (Kurutach et al., 2018; Luo et al., 2018; Nagabandi et al., 2018; Ha & Schmidhuber, 2018; Hafner et al., 2019; Wang & Ba, 2019; Janner et al., 2019), MBRL is able to achieve some level of success. Corneil et al. (2018) and Hafner et al. (2020) also learn a discrete latent representation of the environment in the MBRL framework. As discussed in the introduction, planning on action sequences will fundamentally struggle to scale in robotics.

Our method will make extensive use of a parametric goal-based RL agent to accomplish low-level navigation between states. This area has seen rapid progress recently, largely stemming from the success of Hindsight Experience Replay (HER) (Andrychowicz et al., 2017). Several improvements to HER augment the goal relabeling and sampling strategies to improve performance (Nair et al., 2018; Pong et al., 2018; 2019; Zhao et al., 2019; Pitis et al., 2020). There have also been attempts at incorporating search as inductive biases within the value function (Silver et al., 2016b; Tamar et al., 2016; Farquhar et al., 2017; Racanière et al., 2017; Lee et al., 2018; Srinivas et al., 2018). The focus of this line of work is to improve the low-level policy and is thus orthogonal to our work.

Recent work in Hierarchical RL (HRL) builds upon goal-based RL by learning a high-level parametric manager that feeds goals to the low-level goal-based agent (Dayan & Hinton, 1993; Vezhnevets et al., 2017; Nachum et al., 2018). This can be viewed as a parametric alternative to classical planning, as discussed in the introduction. Recently, Jurgenson et al. (2020); Pertsch et al. (2020) have derived HRL methods that are intimately tied to tree search algorithms. These papers are further connected to a recent trend in the literature wherein classical search methods are combined with parametric control (Savinov et al., 2018a; Eysenbach et al., 2019; Huang et al., 2019; Liu et al., 2019; Yang et al., 2020; Laskin et al., 2020). Several of these articles will be discussed throughout this paper. LEAP (Nasiriany et al., 2019) also considers the problem of proposing sub-goals for a goal-conditioned agent: it uses a VAE (Kingma & Welling, 2013) and does CEM on the prior distribution to form the landmarks. Our method constrains the latent space with temporal reachability between goals, a concept previously explored in Savinov et al. (2018b), and uses latent clustering and graph search rather than sampling-based methods to learn and propose sub-goals.

## 3 BACKGROUND

We consider the problem of Multi-Goal RL under a Markov Decision Process (MDP) that is parameterized by $(S, A, \mathbb{P}, G, \Psi, R, \rho_0)$. $S$ and $A$ are the state and action space. The probability distribution of the initial states is given by $\rho_0(s)$, and $\mathbb{P}(s'|s, a)$ is the transition probability. $\Psi : S \mapsto G$ is a mapping from the state space to the goal space, which assumes that *every* state $s$ can be mapped to a corresponding *achieved* goal $g$. The reward function $R$ can be defined as $R(s, a, s', g) = -\mathbb{1}\{\Psi(s') \neq g\}$. We further assume that each episode has a fixed horizon $T$.

The goal-conditioned policy is a probability distribution $\pi : S \times G \times A \to \mathbb{R}^+$. The policy gives rise to trajectory samples of the form $\tau = \{s_0, a_0, g, s_1, \cdots s_T\}$. The purpose of the policy $\pi$ is to learn how to reach the goals drawn from the goal distribution $p_g$, which means maximizing the cumulative rewards. Together with a discount factor $\gamma \in (0, 1)$, the objective is to maximize $\mathcal{J}(\pi) = \mathbb{E}_{g \sim p_g, \tau \sim \pi(g)}[\sum_{t=0}^{T-1} \gamma^t \cdot R(s_t, a_t, s_{t+1}, g)]$. Q-learning provides a sample-efficient way to optimize the above objective by utilizing off-policy data stored in a replay buffer $B$. $Q(s, a, g)$ estimates the reward-to-go under the current policy $\pi$ conditioned upon the given goal. An additional technique, called Hindsight Experience Replay, or HER (Andrychowicz et al., 2017), uses hindsight relabelling to drastically speed up training. This relabeling crucially relies upon the mapping $\Psi : S \mapsto G$ in the multi-goal MDP setting. We can write the the joint objective of multi-goal Q-learning with HER as minimizing:

$$\min_Q \mathbb{E}_{\substack{\tau \sim B, t \sim \{0 \cdots T-1\} \\ (s_t, a_t, s_{t+1}) \sim \tau \\ k \sim \{t+1 \cdots T\}, g = \Psi(s_k) \\ a' \sim \pi(\cdot|s_{t+1}, g)}} \left( Q(s_t, a_t, g) - \Big( R(s_t, a_t, s_{t+1}, g) + \gamma \cdot Q(s_{t+1}, a', g) \Big) \right)^2 \quad (1)$$

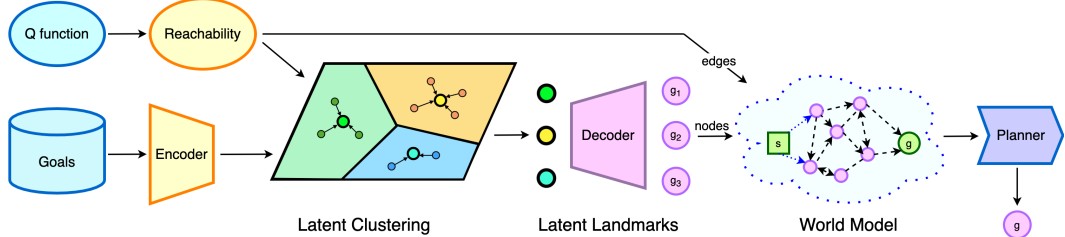

Figure 2: An overview of $L^3P$, which learns a small number of latent landmarks for planning. The main components of our method are: learning reachability estimates (via Q-learning and regression), learning a latent space (via an auto-encoder with reachability constraints), learning latent landmarks (via clustering in the latent space), graph search on the world model and online planning.

## 4    THE $L^3P$ ALGORITHM

Our overall objective in this section is to derive an algorithm that learns a small number of landmarks scattered across goal space in terms of reachability and use those learned landmarks for planning. There are three chief difficulties we must overcome when considering such an algorithm. First, how can we group together goals that are easily reachable from one another? The answer is to embed goals into a latent space, where the latent representation captures some notion of temporal distance between goals – in the sense that goals that would take many timesteps to navigate between are further apart in latent space. Second, we need to find a way to learn a sparse set of landmarks used for planning. Our method performs clustering on the constrained latent space, and decodes the learned centroids as the landmarks we seek. Finally, we need to develop a non-parametric planning algorithm responsible for selecting sequences of landmarks the agent must traverse to accomplish its high-level goal. The proposed online planning algorithm is simple, scalable, and robust.

### 4.1    LEARNING A LATENT SPACE

Let us consider the following question: "How should we go about learning a latent space of goals where the metric reflects reachability?" Suppose we have an auto-encoder (AE) in the agent's goal space, with deterministic encoder $f_E$ and decoder $f_D$. As usual, the reconstruction loss is given by $\mathcal{L}_{rec}(g) = \left\| f_D\big(f_E(g)\big) - g \right\|_2^2$. We want to make sure that the distance between two latent codes would roughly correspond to the number of steps it would take the policy to go from one goal to another. Concretely, for any pair of goals $(g_1, g_2)$, we optimize the following loss:

$$\mathcal{L}_{latent}(g_1, g_2) = \left( \left\| f_E(g_1) - f_E(g_2) \right\|_2^2 - \frac{1}{2}\Big( V(g_1, g_2) + V(g_2, g_1) \Big) \right)^2 \tag{2}$$

Where $V : G \times G \to \mathbb{R}^+$ is a mapping that estimates how many steps it would take the policy $\pi$ to go from one goal to another goal on average. By adding this constraint and solving a joint optimization $\mathcal{L}_{rec} + \lambda \cdot \mathcal{L}_{latent}$, the encoding-decoding mapping can no longer be arbitrary, giving more structure to the latent space. Goals that are close by in terms of reachability will be naturally clustered in the latent space, and interpolations between latent codes will lead to meaningful results.

Of course, the constraint in Equation 2 is quite meaningless if we do not have a way to estimate the mapping $V$. We will proceed towards this objective by noting the following interesting connection between multi-goal Q-functions and reachability. In the multi-goal RL framework considered in the background section, the reward is binary in nature. The agent receives a reward of $-1$ until it reaches the goal, and then $0$ when it reaches the desired goal. In this setting, the Q-function is implicitly estimating *the number of steps* it takes to reach the goal $g$ from the current state $s$ *after* the action $a$ is taken. Denote this quantity as $D(s, a, g)$, the Q-function can be re-written as:

$$Q(s, a, g) = \sum_{t=0}^{D(s,a,g)-1} \gamma^t \cdot (-1) + \sum_{t=D(s,a,g)}^{T-1} \gamma^t \cdot 0 = -\frac{1 - \gamma^{D(s,a,g)}}{1 - \gamma} \tag{3}$$

Choosing to parameterize Q-functions in this way disentangles the effect of $\gamma$ on multi-goal Q-learning. It also provides us with access the direct distance estimation function $D(s, a, g)$. We note

that this *distance* is not a mathematical distance in the sense of a metric. Instead, we use the word *distance* to refer to the number of steps the policy $\pi$ needs to take in the environment.

Given our tractable estimate of $D$, it is now a straightforward matter to estimate the desired quantity $V$, which approximates how many steps it takes the policy to transition between goals. To get the desired estimate, we regress $V$ towards $D$ as follows

$$\min_{V} \mathbb{E}_{\substack{\tau \sim B, t \sim \{0 \cdots T-1\} \\ (s_t, a_t, s_{t+1}) \sim \tau \\ k \sim \{t+1 \cdots T\}}} \left( D\big(s_t, a_t, \Psi(s_k)\big) - V\big(\Psi(s_{t+1}), \Psi(s_k)\big) \right)^2 \tag{4}$$

where $\Psi$ is given by the environment to map the states to the goal space. One crucial detail is the use of $\Psi(s_{t+1})$ rather than $\Psi(s_t)$ in the inputs to $V$. This is due to the fact that $D : S \times A \times G \to \mathbb{R}$ outputs the number of steps to go *after* an action is taken, when the state has transitioned into $s_{t+1}$. The objective above provides an unbiased estimate of the average number of steps between two goals.

The estimates $D$ and $V$ will prove useful beyond helping to optimize the auto-encoder in Equation 2. They will prove essential in weighting and planning over latent landmark nodes in Section 4.3.

## 4.2 Learning Latent Landmarks

Planning on a graph can be expensive, as the number of edges can grow quadratically with the number of nodes. To battle this issue in scalability, we use the constrained latent space to learn a sparse set of landmarks. A landmark can be thought of as a waypoint that the agent can pass through enroute to achieve a desired goal. Ideally, *goals that are easily reachable from one another should be grouped to form one single landmark*. Since our latent representation captures the temporal reachability between goals, this can be achieved by doing clustering in the latent space. The cluster centroids, when decoded from the decoder, will be precisely the latent landmarks we are seeking.

Clustering proceeds as follows. For $N$ clusters to be learned, we define a mixture of Gaussians in the latent space with $N$ trainable latent centroids, $\{\mathbf{c}_1 \cdots \mathbf{c}_N\}$, and a shared trainable variance vector $\boldsymbol{\sigma}$. We maximize the evidence lower bound (ELBO) with a uniform prior $p(\mathbf{c})$:

$$\log p\Big(z = f_E(g)\Big) \geq \mathbb{E}_{q(\mathbf{c}|z)}\Big[ \log p(z \mid \mathbf{c}) \Big] - D_{KL}\Big(q(\mathbf{c} \mid z) \parallel p(\mathbf{c})\Big) \tag{5}$$

Ideally, we would like each batch of data given to the latent clustering model to be representative of the whole replay buffer, such that the centroids will quickly learn to scatter out. To this end, we propose to use the Greedy Latent Sparsification (GLS) algorithm (Algorithm 2 in the Appendix) on each batch of data sampled from the replay before taking a gradient step with the batch. GLS is inspired by kmeans++ (Arthur & Vassilvitskii, 2007), with several key differences: this sparsification process is used for both training and initialization, it uses a neural metric for determining the distance between data points, and that it is compatible with mini-batch-style gradient-based training.

## 4.3 Planning with Latent Landmarks

Having derived a latent encoding algorithm and an algorithm for learning latent landmarks, we at last turn our attention to planning. While prior works simply solve for the shortest path, we employ a *soft* version of the Floyd algorithm, where the soft relaxation operations can be seen as a soft value iteration procedure (see Equation 7 in the Appendix).

To construct a weight matrix that at first provides a raw distance estimate between latent landmarks, we begin by decoding the learned centroids in the latent space into the nodes in the graph $\{f_D(\mathbf{c}_1) \cdots f_D(\mathbf{c}_N)\}$. To build the graph, we add two edges directed in reverse orders for every pair of latent landmarks. For instance, for an edge going from $f_D(\mathbf{c}_i)$ to $f_D(\mathbf{c}_j)$, the weight on that edge is $-V(f_D(\mathbf{c}_i), f_D(\mathbf{c}_j))$. Notice that the distances are negated to be negative. At the start of an episode, the agent receives a goal $g$, and we construct the following matrix of size $(N+1) \times (N+1)$:

$$W = \begin{pmatrix} 0 & \cdots & -V(f_D(\mathbf{c}_1), f_D(\mathbf{c}_N)) & -V(f_D(\mathbf{c}_1), g) \\ \vdots & \ddots & \vdots & \vdots \\ -V(f_D(\mathbf{c}_N), f_D(\mathbf{c}_1)) & \cdots & 0 & -V(f_D(\mathbf{c}_N), g) \\ -\infty & \cdots & -\infty & 0 \end{pmatrix} \tag{6}$$

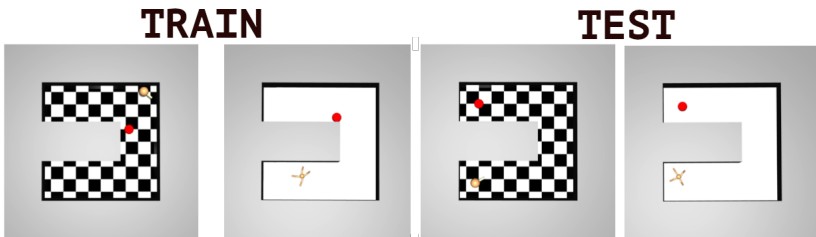

Figure 3: For both Point and Ant, during training, the initialization state distribution and the goal proposal distribution are *uniform* around the maze. During test time, the agent is asked to traverse the longest path in the maze. The success rate on the test environment is reported in Figure 4. This environment demonstrates $L^3P$'s ability to generalize to longer horizon goals during test time.

---

**Algorithm 1** Online Planning in $L^3P$

**Given**: Environment env, initial state $s$, goal $g$.

1: Cnt = 0. SubG = None.
2: Solve for $\boldsymbol{d_{c \to g}}$ with graph search.
3: **for** $t = 1$ to $T$ **do** ▷ One episode
4:    **if** Cnt $\geq 1.0$ **then**
5:       Cnt = Cnt $-$ 1
6:    **else** ▷ We do **not** re-plan at every step
7:       Calculate $\boldsymbol{d_{s \to c}}$.
8:       $\boldsymbol{d} \leftarrow \boldsymbol{d_{s \to c}} + \boldsymbol{d_{c \to g}}$
9:       **if** SubG $\neq$ None **then**
10:          $\boldsymbol{d}[\text{SubG}] \leftarrow -\infty$
11:       **end if** ▷ Remove the **immediate previous landmark**
12:       SubG, Cnt $\leftarrow \arg\max \boldsymbol{d}, -\max \boldsymbol{d}$
13:    **end if**
14:    $a \sim \pi(s, \text{SubG}); s \leftarrow \text{env.step(a)}.$
15: **end for**

For online planning, when the agent receives a goal at the start of an episode, we use graph search to solve for $\boldsymbol{d_{c \to g}}$ (which is fixed throughout an episode). For an observation state $s$, the algorithm calculates $\boldsymbol{d_{s \to c}}$:

$$\boldsymbol{d_{s \to c}} = \begin{pmatrix} -D\big(s, \pi(s, f_D(\mathbf{c}_1)), f_D(\mathbf{c}_1)\big) \\ \vdots \\ -D\big(s, \pi(s, f_D(\mathbf{c}_N)), f_D(\mathbf{c}_N)\big) \\ -D\big(s, \pi(s, g), g\big) \end{pmatrix}$$

The chosen landmark is subgoal $\leftarrow \arg\max(\boldsymbol{d_{s \to c}} + \boldsymbol{d_{c \to g}})$. To further provide temporal abstraction and robustness, the agent will be asked to consistently pursue subgoal for $-\boldsymbol{d_{s \to c}}[\text{subgoal}]$ number of steps, which is *how many steps it thinks it will need*. The proposed goal does **not** change during this period.

The algorithm makes sure that the agent does not re-plan at every step, and this mechanism for temporal abstraction is crucial to its robustness. After this many steps, the agent will decide on the next landmark to pursue by re-calculating $\boldsymbol{d_{s \to c}}$, but the *immediate* previous landmark will not be considered as a candidate landmark. The reason is that, if the agent has failed to reach a self-proposed landmark within the reachability limit it has set for itself, then the agent should try something new for the immediate next goal rather than stick to the immediate previous landmark for another round. We have found that this simple algorithm helps the agent avoid getting stuck and improves the overall robustness of the agent.

In summary, we have derived an algorithm that learns a sparse set of latent landmarks scattered across goal space in terms of reachability, and uses those learned landmarks for robust planning.

## 5 EXPERIMENTS AND EVALUATION

We investigate the impact of $L^3P$ in a variety of robotic manipulation and navigation environments. These include standard benchmarks such as Fetch-PickAndPlace, and more difficult environments such as AntMaze-Hard and Place-Inside-Box that have been engineered to require test-time generalization. Videos of our algorithm in action are available here: `https://sites.google.com/view/latent-landmarks/`.

### 5.1 BASELINES

We compare our method with a variety of baselines. HER (Andrychowicz et al., 2017) is a model-free RL algorithm. SORB (Eysenbach et al., 2019) is a method that combines RL and graph search by using the entire replay buffer. Mapping State Space (MSS Huang et al. 2019) reduces the number

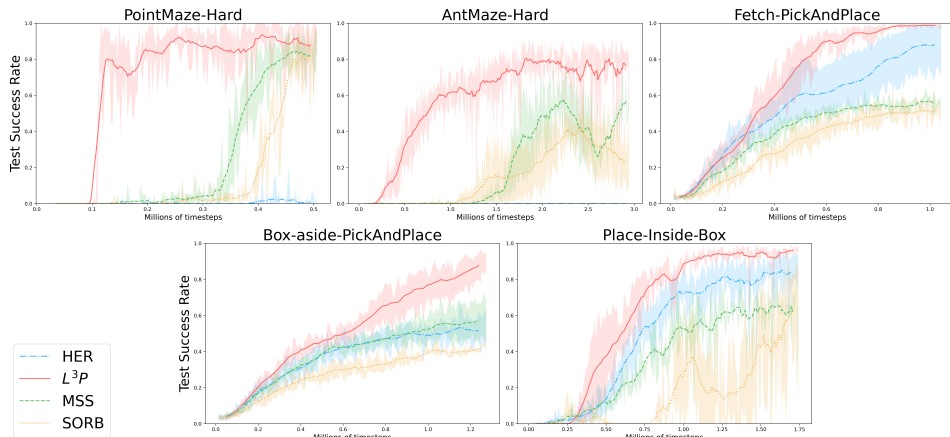

Figure 4: Test time success rate vs. total number of timesteps, on maze and robotic manipulation environments. During test time, new more difficult goals are selected. $L^3P$ shows more robust generalization much more quickly than other methods. For every environment except PointmMaze, $L^3P$ is the only algorithm that consistently solves the task.

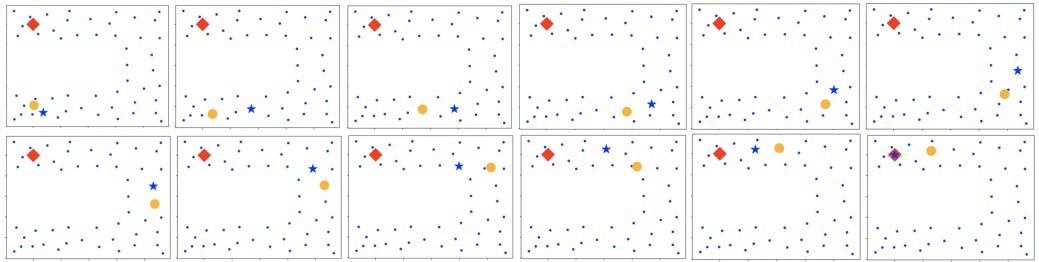

Figure 5: Visualizing planning on AntMaze at test time. Read images from upper left to bottom right. The blue dots are the learned latent landmarks decoded from the latent centroids. The orange dot represents the ant's present location in the maze. The red dot is the final goal that the agent needs to reach. At each step, the blue star indicates the landmark chosen by our planning algorithm. Whereas MSS and SORB sample 400 and hundreds of thousands of landmarks (respectively), our method obtains a lean graph that only contain 50 landmarks. $L^3P$ is the only method capable of achieving over 80% test success rate on AntMaze-Hard within 3M timesteps.

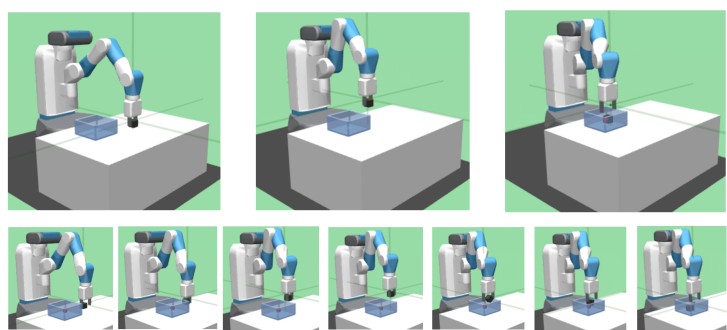

Figure 6: We consider two environments involving a fetch robot, a block, and a box. In Box-aside-PickAndPlace, the fetch must learn to pick and place the block while avoiding collision with the box. In Place-Inside-Box, the fetch must pick the block and place it inside the box. We visualize the fetch states corresponding to learned landmarks in the second row of images.

of vertices by sub-sampling the replay buffer. $L^3P$, SORB, and MSS all use the same hindsight relabelling strategy proposed in HER. All of the domains are continuous control tasks, so we adopt DDPG (Lillicrap et al., 2015) as the learning algorithm for the low-level actor.

## 5.2 GENERALIZATION TO LONGER HORIZONS

The PointMaze-Hard and AntMaze-Hard environments introduced in Figure 5 are designed to test an agent's ability to generalize to longer horizons. While PointMaze and AntMaze have been previously used in Duan et al. (2016); Huang et al. (2019); Pitis et al. (2020), we make slight changes to those environments in order to increase their difficulty. We use a short, 200-timestep time horizon during training and a $\rho_0$ that is uniform in the maze. At test time, we always initialize the agent on one end of the maze, and set the goal on the other end. The horizon of the test environment is 500 steps. Crucially, no prior knowledge on the shape of the maze is given to the agent. We also set a much stricter threshold for determining whether an agent has reached the goal. In Figure 4, we see $L^3P$ is the only algorithm capable of solving AntMaze-Hard consistently.

We observe an interesting trend where the success rates for other graph search methods crash and then slowly recover after making some initial progress. We postulate this occurs because methods that are based on using the entire replay or sub-sampling the replay for landmark selection will struggle as the buffer size increases. In contrast to these methods, $L^3P$ does not exhibit such undesirable instability. The online planning algorithm in $L^3P$, which effectively leverages temporal abstraction to improve robustness, also contributes to the asymptotic success rate. The result convincingly shows that, at least on the navigation tasks considered, $L^3P$ is most effective at taking advantage of the problem's inherent graph structure, and that learning latent landmarks is significantly more sample efficient and scalable than directly using or sub-sampling the replay buffer to build the graph.

## 5.3 ROBOTIC MANIPULATION TASKS

We also benchmark challenging robotic manipulations tasks with a Fetch robot introduced in Plappert et al. (2018); Andrychowicz et al. (2017). In Figure 6, we introduce two pick and place tasks involving a box on a table. In the Place-Inside-Box environment, we design a simple curriculum to cope with the difficulty of the task. During training, the goal distribution has 80% regular pick-and-place goals, enabling the agent to first learn how to fetch in general. Meanwhile, only 20% of the goals are inside the box, which is the harder part of the task. During testing, we evaluate the ability of the agent to pick the object from the table and place it inside the box. Our method achieves dominant performance in both learning speed and test-time generalization. We note that on those manipulation tasks considered, many prior planning methods *hurt* the performance of the model-free agent. Our method is the only one that is able to help the model-free agent learn faster and generalize better.

## 5.4 UNDERSTANDING MODEL CHOICES IN $L^3P$

We investigate $L^3P$'s sensitivity to different hyper-parameters and design choices via a set of ablation studies. More specifically, we study how the following factors affect the performance of $L^3P$: number of latent landmarks, the choice of (online) planning algorithms, the choice of graph search algorithms, and edge cutoff threshold in graph search (a key hyper-parameter in the search module).

The first question we try to understand is whether $L^3P$ is robust to the number of latent landmarks. In contrast to prior methods, $L^3P$ is able to *learn* the landmarks used for graph search from the agent's own experience. We vary the number of learned landmarks in the challenging AntMaze-Hard environment, and we find that $L^3P$ is robust against a decreasing number of landmarks. This is expected, because the landmarks in the latent space of $L^3P$ will try to be equally scattered across the goal space according to the reachability metric. As the number of landmarks decreases, the learning procedure will automatically push the landmarks to be further away from one another.

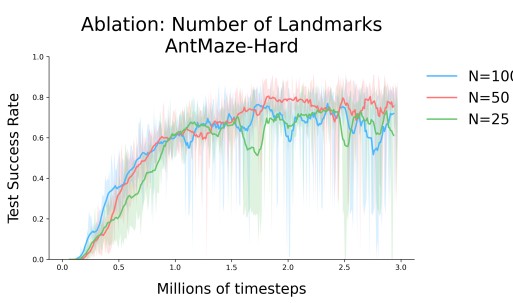

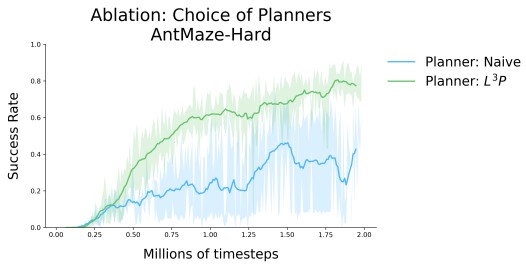

A key component in $L^3P$ is the online planning algorithm described in Algorithm 1. We find this algorithm to bear special importance to the good performance of $L^3P$. Our planning algorithm in $L^3P$ can take advantage of the temporal abstraction provided by the graph-structured world model. It does not re-plan at every step, but instead uses the reachability estimates to dynamically decide when to re-plan, striking a balance between adaptability and consistency in planning. This planner is also more tolerant of errors: it removes the immediate previous landmark when it re-plans, so that the agent will be less prone to getting stuck. A naive planner, on the other hand, simply re-calculates the shortest path at every step. The curve on the left shows that this planning algorithm is crucial to the success of $L^3P$.

The particular choice of graph search seems to have a small effect on the stability of learning. As explained Section 4.3 and Appendix A.2, we find that while employing the Floyd algorithm for graph search, a *soft* operation for relaxation leads to better stability during training. On the right, we show that a hard version of relaxation helps the agent take off faster but suffers from greater instability during policy improvement. The likely reason is that neural distance estimates are not entirely accurate, and in the presence of occasional bad edges, $\mathrm{softmax}$ in Equation 7 improves robustness. We therefore use soft relaxation in our graph search module.

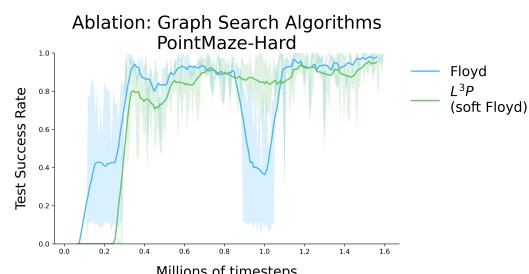

One of the most important hyper-parameters when combining RL with graph search is $d\_max$, the clipping threshold for the edges on the graph (Savinov et al., 2018a; Eysenbach et al., 2019; Huang et al., 2019; Laskin et al., 2020). The motivation for introducing this commonly used hyper-parameter is two-fold. Firstly, we only trust distance estimates when they are *local*. Secondly, we want the graph search module to produce sub-goals that are *nearby*. The $d\_max$ value determines the maximum distance for each edge on the graph and masks out longer

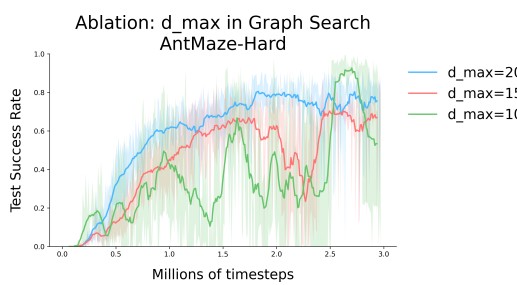

edges. One weakness of our current approach is that it is still quite sensitive to this hyper-parameter; a small change to $d\_max$ can have considerable impacts on learning. As this weakness is common to this class of approaches, we believe that further research is required to find other ways of encouraging search to be *local*. See Appendix A.2 for more details on implementing this clipping threshold.

## 6  CLOSING REMARKS

In this work, we introduce a way of learning graph-structured world models that endow agents with the ability to do temporally extended reasoning. The algorithm, $L^3P$, learns a set of latent landmarks scattered across the goal space to enable scalable planning. We demonstrate that $L^3P$ achieves significantly better sample efficiency, higher asymptotic performance, and better generalization on a range of challenging robotic navigation and manipulation tasks. We hope that this work inspires more research in the direction of combining deep RL with classical planning.

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

# A    APPENDIX

## A.1    GREEDY LATENT SPARSIFICATION

---
**Algorithm 2** Greedy Latent Sparsification (GLS) for Latent Cluster Training
---
**Given**: Replay Buffer $B$, Encoder $f_E$.
**Initialize**: `LatentEmbeds` $= \{\}$.                  ▷ Set of embeddings selected.
 1: Sample $K$ achieved goals from $B$.
 2: Sample $k \sim \{0 \cdots K - 1\}$.
 3: `dist` $= [\|f_E(g_1) - f_E(g_k)\|_2^2, \cdots, \|f_E(g_K) - f_E(g_k)\|_2^2]$
 4: **for** i = 1 to $M$ **do**                              ▷ Sub-sampling
 5:     $k \leftarrow \arg\max \texttt{dist}[k]$
 6:     Add $f_E(g_k)$ to `LatentEmbeds`.
 7:     `NEWdist` $= [\|f_E(g_1) - f_E(g_k)\|_2^2, \cdots, \|f_E(g_K) - f_E(g_k)\|_2^2]$
 8:     `dist = ElementwiseMin(dist,NEWdist)`
 9: **end for**
10: Optimize equation 5 on `LatentEmbeds`.
---

The Greedy Latent Sparsification (GLS) algorithm sub-samples a large batch by sparsification. GLS first randomly selects a latent embedding from the batch, and then greedily chooses the next embedding that is furthest away from already selected embeddings. After collecting some *warm-up trajectories* before planning starts (see Table 2) during training, we first use GLS to initialize the latent centroids, and then continue to use it to sample the batches used to train the latent clusters. As mentioned in Section 4.2, GLS is strongly inspired by Arthur & Vassilvitskii (2007), and this type of approach is known to improve clustering.

## A.2    GRAPH SEARCH VIA SOFT VALUE ITERATIONS

In this paper, we employ a soft version of Floyd algorithm, which we find to empirically work well. Rather than simply using the $\min$ operation to do relaxation, the soft value iteration procedure uses a $soft \min$ operation when doing an update (note that, since we negated the distances to be negative in the weight matrix of the graph, which is Equation 6, the operations we use are actually $\max$ and $\mathrm{softmax}$). The reason is that neural distances can be inconsistent and inaccurate at times, and using a soft operation makes the whole procedure more robust. More concretely, we repeat the following update on the weight matrix for $S$ steps with temperature $\beta$:

$$w_{i,j} \leftarrow \sum_{k=1}^{N+1} \frac{\exp \dfrac{1}{\beta}(w_{i,k} + w_{k,j})}{\sum_{k'=1}^{N+1} \exp \dfrac{1}{\beta}(w_{i,k'} + w_{k',j})} \Big( w_{i,k} + w_{k,j} \Big) \tag{7}$$

Following the practice in Eysenbach et al. (2019); Huang et al. (2019), we do the following initialization to the matrix in Equation 6: for entries smaller than the negative of $d\_max$, we penalize the entry by adding $-\infty$ to it (in this paper, we use $-10^6$ as the $-\infty$ value). The essential idea is that we only trust a neural estimate when it is *local*, and we rely on graph search to solve for *global*, longer-horizon distances. The $-\infty$ penalty effectively masks out those entries with large negative values in the $\mathrm{softmax}$ operation above. If we replace $\mathrm{softmax}$ with a hard $\max$, we recover the original update in Floyd algorithm; we can interpolate between a hard Floyd and a soft Floyd by tuning the temperature $\beta$.

## A.3    HYPER-PARAMETERS

Table 1 lists the common hyper-parameters across all environments. Table 2 lists the hyper-parameters that differ across the environments.

Table 1: Hyper-parameters in Common

| Parameter | Value |
|---|---|
| *DDPG* | |
|     optimizer | Adam (Kingma & Ba, 2014) |
|     number of hidden layers (all networks) | 3 |
|     number of hidden units per layer | 256 |
|     nonlinearity | ReLU |
|     polyak for target network ($\tau$) | 0.995 |
|     target update interval | 10 |
|     ratio between env vs optimization steps | 2 |
|     Random action probability | 0.2 |
|     Initial random trajs per worker | 100 |
|     Hindsight relabelling ratio | 0.85 |
| *Latent Landmarks & Auto-encoder* | |
|     number of hidden layers | 2 |
|     number of hidden units per layer | 128 |
|     nonlinearity | ReLU |
|     embedding size | 16 |
|     $\lambda$ for reachability constraint loss | 1.0 |
|     learning rate | 3e-4 |
| *Graph Search* | |
|     probability of using search during train | 0.5 |
|     $S$ (number of soft value iterations) | 20 |
|     $\beta$ (temperature) | 1.1 |

Table 2: Hyper-parameters for Each Environment

| | Point-Maze | Ant-Maze | Fetch tasks |
|---|---|---|---|
| *DDPG* | | | |
| Learning rate | 2e-4 | 2e-4 | 1e-3 |
| Number of workers | 1 | 3 | 12 |
| Batch size | 512 | 1024 | 1024 |
| Action L2 | 0.5 | 0.05 | 0.01 |
| Gamma | 0.98 | 0.98 | 0.99 |
| Action noise | 0.2 | 0.2 | 0.1 |
| Hindsight relabelling range | 80 | 100 | 50 |
| *Latent Landmarks & Auto-encoder* | | | |
| Number of latent landmarks | 50 | 50 | 80 |
| Number of *warm-up trajectories* | 500 | 500 | 6000 |
| Batch size | 256 | 256 | 150 |
| *Graph Search* | | | |
| $d\_max$ (clipping threshold for distances) | 20.0 | 20.0 | 15.0 |
| Random landmarks added during train | 150 | 150 | 20 |

- We find that having a centralized replay for all parallel workers is significantly more sample efficient than having separate replays for each worker and simply averaging the gradients across workers.

- For Ant-Maze environment, we do grad norm clipping by a value of $15.0$ for all networks. For Fetch tasks, we normalize the inputs by running means and standard deviations per input dimensions.

- Since $L^3P$ is able to decompose a long-horizon goal into many short-horizon goals, we shorten the range of future steps where we do hindsight relabelling; as a result, the agent can focus its optimization effort on more immediate goals. This corresponds to the hyper-parameter: Hindsight relabelling range.

- During training, we collect $50\%$ of the data without the planning module, and the other $50\%$ of the data with planning. This corresponds to the hyper-parameter: probability of using search during train.

- At train time, to encourage exploration during planning, we temporarily add a small number of random landmarks from GLS (Algorithm 2) to the existing latent landmarks. A new set of random landmarks is selected for each episode before graph search starts (Algorithm 1). This corresponds to the hyper-parameter: Random landmarks added during train.

- We find that collecting a certain number of *warm-up trajectories* for every worker before the planning procedure starts (during training) and before GLS (Algorithm 2) is used for initialization to help improve the planning results. This corresponds to the hyper-parameter: Number of *warm-up trajectories*.

