# OpenReview forum: "World Model as a Graph: Learning Latent Landmarks for Planning"
_ICLR.cc/2021/Conference — Reject_

### Official Review · AnonReviewer2 · 2020-10-28
**A better graph-based planning algorithm for goal-conditioned RL, but the presentation is not very clear.**

**Rating:** 6
**Confidence:** 4

**Review:**

This paper generalizes the previous graph-based planning algorithm for goal-conditioned RL algorithms by learning a latent metric space and building the graph by clustering. The experiments show that their approach outperforms both the HER and previous MSS/SORB baselines.

Strong points:
1. The paper achieves good performance on the challenging AntMaze and PickAndPlace environments.
2. The clustering methods to find latent landmarks are novel in this setting.

Weak points:
1. The novelty of the paper is limited. Planning on the graph or learning a latent distance embedding for planning are all existing ideas. The combination is straightforward. The robustness of soft-min is well known.
2. I am confused about why the authors' approach is better than MSS or SORB. It's not obvious that a distance-based clustering will perform significantly better than sampled landmarks (like farthest point sampling). I guess the reason is that clustering helps to avoid the landmarks near the wall (which often appears in the FP sampled methods). Those states are challenging as it has a higher possibility for the ant to collide with the wall. Can the author give some words about this? Moreover, the paper is not clear about the contributions of each component in the improvements. For example, no table or figure shows the benefits of the soft value iteration over the hard version. The author shall provide more ablation studies and explanations.
3. Although the paper presents the overall idea well, there is still a lot of room for improvement in writing. For example, I suggest that the author replaces the hyphen with the comma in the abstract.

~~~Based on the current presentation, the lack of ablation study, and the limited novelty, I think this paper is not good enough to be accepted. ~~~

Here are some questions:
1. Can we build graphs by sampling goal states and cluster them based on the metric space defined by the Q networks? What's the performance of this approach? Will it be worse? Why?
2. Can a sample-based approach work in non-navigation environments? Motion planning algorithms can solve the high-level part of both pick&place or ant maze problem once the geometric model is known. If so, why do we need the expensive RL algorithm?


---

After author response：

The authors have improved the presentation and added the necessary ablation studies. I appreciate the authors' effort. I am glad to raise the score to reflect the changes. I hope this work will inspire future research on hierarchical planning algorithms.

---

> ### Author Response · Authors · 2020-11-23
> **Thank you for your review!**
>
> Thank you for your review! We have updated the draft to address many of your concerns.
>
> Regarding the confusion surrounding what specifically made the algorithm work better, we have added new ablations and thorough discussions in our updated draft. Besides learning a sparse set of landmarks, we find our online planning algorithm (Algorithm 1) to be crucial to the performance of our method, and we have added an ablation on the choice of planners. Regrading your point about collisions, we have indeed observed that our method seems to be better at avoiding collision in the AntMaze environment. We think the likely reason is that fewer goals are reachable when the ant is near collision with the wall, and thus reachability-based latent clustering techniques tend to avoid landmarks near the walls. Another ablation we added was the soft versus hard min in graph search. As you have pointed out, the robustness of soft-min is well known; the simplest way to explain our graph search module is that we replace the hard min in Floyd algorithm with a soft min. This is discussed both in the new ablation studies and the appendix. We agree that those discussions are important for understanding why exactly the algorithm is doing better than MSS and other graph search algorithms.
>
> Regarding your question about sampling based goal states, we note that one of the primary contributions in our method is that it no longer needs to subsample the replay buffer for landmarks anymore. We have compared against MSS, which uses sampling based landmark generation. Our method is able to perform better mainly because subsampling the replay is not particularly scalable as the replay grows large, and that learning a smaller set of sparse landmarks enables the agent to better leverage temporal abstraction.
>
> Regarding your point on motion planning: we will be happy to run motion planning algorithms as a baseline if there have been known methods for motion planning to scale to environments such as AntMaze and PickAndPlace without topology or geometry being given the agent as prior inputs (for a fair comparison). A major challenge in solving environments like AntMaze is that the agent has to (either explicitly or implicitly) find out the topology of the environment on its own via exploration.
>
> In our updated draft, we have also tried to improve the overall clarity of the paper by providing new Figures (1 and 2) and new ablations, and also by putting substantial effort into re-writing the algorithm and experiment sections of the paper. We hope that the new draft clarifies this work’s main contributions and addresses the points you raised.

---

### Official Review · AnonReviewer4 · 2020-10-28
**Introduces new method for state abstraction on a latent space with compelling experiments. Writing unclear in some places.**

**Rating:** 7
**Confidence:** 3

**Review:**

This paper proposes an approach for automatically learning state abstraction on a RL problem, which can then be used for temporally extended planning using a search algorithm. The main contribution is introducing the concept of latent *landmarks*, a clustering of low dimensional state embeddings. Landmarks are defined on a latent space wherein the distance between two latents code is small if their corresponding high-dimensional states can reach each other in few environment steps. The paper proposes to cluster latent states, so that each cluster contains states that are easy to reach from each other; the landmarks correspond to the centers of these clusters. An algorithm for automatically learning this clustering is introduced in the paper.  With landmarks in hand, the paper proposes a soft-value iteration method to compute shortest path distances to the problem's goal. The approach is evaluated in a variety of domains, and compares favorably with recent state-of-the-art methods.

In general, I found the paper interesting and the key ideas intuitive and sensible. The paper is reasonably well-written, but there are some important points that are unclear (see below). The experimental section compares with very recent algorithms for planning over learned value functions (SORB and MSS), and the results on five continuous control task show substantial improvement over these methods.

In terms of clarification, I have a few questions for the authors:
- With regards to the embedding used and $\Psi$:
  - The embedding operates over goals. Are these the goals of the problem? For example, in the AntMaze problem (Fig. 4), is $G = \{ \textit{red-square} \} $? If so, doesn't this mean that  $\forall s \Psi(s) = \textit{red-square}$, and thus $V(\Psi(s), \Psi(s')) = 0$ for all pairs of states?
  - More reasonably, $V$ could be defined over states, so that $V(s_1, s_2)$ indicates the number of steps between $s_1$ and $s_2$. This can be achieved under the current formulation with $G = S$ and $\Psi(s) = s$. Then $D(s_t, a, \Psi(s_t))$ represents the number of steps between $s_{t+1}$ and $s_t$. Is this the case for most experiments? No details of the $\Psi$ function used in the experiments was given, so I find this point somewhat confusing.
  - If this is not the case, then I think it's important to explain where does this mapping function comes from, since it essentially provides some reward shaping for the problem.

- Can you explain what is the motivation for using Algorithm 2? A common approach is to use Djikstra on the graph representation. Is there a reason why this can't be done (or is worse) in this case?

- I don't fully understand Algorithm 1 to create the batch for Eq. (5). Can you explain this algorithm in more details? For example, the definition of $\texttt{dist}$ is ambiguous; is it pairwise distance between all sampled goals? distance between $g_1$ and all the others? something else? In general, I'm having quite a hard time figuring out what this is doing.

In light of above, I think the clarity of the paper can be improved in many places. But, overall, I'm positive towards this work and I think it is a nice contribution, particularly since I'm not aware of other work creating explicit low-dimensional landmarks to be used for search. That being said, one exception that is missing from the literature review is [1], which creates a discrete latent representation of the environment and solves it using prioritized sweeping.

[1] Corneil, Dane, Wulfram Gerstner, and Johanni Brea. "Efficient Model-Based Deep Reinforcement Learning with Variational State Tabulation." International Conference on Machine Learning. 2018.

---

> ### Author Response · Authors · 2020-11-23
> **Thank you for your review!**
>
> Thank you for your review and the reference you provided! We have updated the paper to include the reference.
>
> With regards to the \Psi function used: we note that \Psi function is explained in the background section as a crucial assumption for hindsight relabelling techniques like HER, and here we are simply writing it out explicitly. HER assumes that we can map every state the agent has seen to its corresponding achieved goal. The goal space defines the elements in an environment which we care about. We refer to the background section and the paragraph in the related work section about goal-conditioned RL for further explanations on this function.
>
> We have included an ablation study where we test against exact shortest path methods like Floyd and Dijkstra. The simplest way to explain this soft-value-iteration component is that we are using a variant of Floyd algorithm for graph search, except that we replace the hard min with a soft min (details see Appendix A.2). We see in the ablation study that indeed the change does make a difference, but the online planning algorithm we have used is much more important.
>
> We have expanded Appendix A.1 to explain the GLS algorithm in more details. The dist vector maintains the minimum distances between already selected embeddings and every embedding in the batch. A simple explanation is that it is a variant of the kmeans++ strategy, which is well known to improve clustering-based approaches.
>
> Regarding the overall clarity of the paper, we have provided new Figures (1 and 2), and also put substantial effort into re-writing the algorithm section of the paper. We hope that this helps to clarify some of the confusion about the exact implementation details of the algorithm. We agree that a good high-level intuition was missing from the presentation before. We hope these rewrites make the general flow of the algorithm more lucid.

---

### Official Review · AnonReviewer3 · 2020-10-28

**Rating:** 5
**Confidence:** 4

**Review:**

#### Summary:
This paper presents a method for learning a sparse set of latent subgoal states during training. Using a goal-conditioned policy and the latent states, a simple planning algorithm that performs soft value iteration between the subgoal states is proposed to facilitate within-dataset generalization. The proposed method outperforms competing approaches on multiple simulated navigation and robotics tasks.

##### Pros:
- The results seem strong.
- The method is straightforward and nice, and it seems to work more efficiently than previous methods.

##### Cons:
- Overall, the paper is incomplete. It’s very difficult to understand the whole system, how training proceeds, what the hyperparameters are, etc. There is very little detail about how this all works. The specific components of the latent landmark learning and the planning are mostly fleshed out (albeit with some missing details still) but the rest of the setup is mostly not described. How are the losses combined? Are there multiple learners or does the DDPG agent also do the latent space learning? How is training done? What are the hyperparameters used?
- The generalization claims are a bit oversold as the generalization achieved is weak. In particular, the method requires that the test goal distribution have sufficient overlap with the train goal distribution for latent landmarks to be created near the desired test goals.
- Are the experiment domains created here or existing work? If the former, more details are required. If the latter, citations are required.
- Given the multiple changes with respect to prior work, additional ablations and experiments are needed to better understand why this method is performing well, and what contributions are important.

#### Decision:
Overall, I find it too hard to know exactly what is happening in this work. It would be very difficult, if not impossible, to reproduce this work from this paper alone. Without sufficient details about the system, it’s hard to evaluate, which means that I default to a reject, especially in conjunction with the other weaknesses listed above.

#### Questions:
1. What is the goal space G? Is it the state space? This should be defined.
2. What exactly is \Psi? Is it learned or is it given? This should be explained better.
3. How is the autoencoder for the latent space learned within the overall system?
4. The connection between D and Q is poorly explained. From the text, it reads as if you get Q from D. However, it makes more sense if the DDPG agent estimates Q and then you use eq(3) to compute D from Q. Which is it?
5. Are both V and D really necessary? It seems like they encode the same thing, essentially.
6. Why do you use soft value iteration? Is it a good choice? Is there a reason for this choice? Would other choices be better or worse? It seems likely that the need for the dist_max penalty is due to the use of the soft value iteration with a poorly set soft value iteration temperature. Is this true?
7. Are the domains from Duan et al 2016 or some other existing work? Or did you create them from scratch (unlikely). Why are they labeled “-hard”? What is hard about them? These need to be cited and described. You have tons of space in the appendix to use freely.
8. In Figure 3, is the x axis training timesteps?
9. Figure 5 caption says the agent has learned landmarks that help avoid collision with the box. Where are these? Is there a figure? Any sort of information on these? Wouldn’t any landmark away from the box accomplish this?
10. For Figure 6, why is choosing the maximum distance bound important? Why would the planning algorithm choose subgoals that are near the max distance bound given that the planner is finding the argmin subgoal? It seems likely that this hacky distance bound is unnecessary.
11. The “planning algorithm” seems trivial. Am I missing something? Given the latent landmarks and the estimates for $d_{c \rightarrow g}$, the planner just gets the next estimated subgoal and the distance to it, executes the goal-conditioned policy for that subgoal for that many steps, crosses out that subgoal, and then goes to the next subgoal. Is it non-trivial because $d_{s \rightarrow c}$ gets updated after each step (not included in the pseudocode)?
12. Is the GLS algorithm a contribution of this work or something from a previous work? Again, it's neither described sufficiently nor cited sufficiently.

#### Comments:
- The introduction uses unnecessary hyperbole and insufficient citations to make its points. It comes off as less compelling as a result.
- Section 3 is called Background but really it contains the preliminaries and definitions of this work. It should not be called Background.
- Should the variance vector of the centroids appear in equation (5)? It is mentioned once and then never discussed again. What values of it are learned? Is it important?
- Similarly, the temperature for the soft value iteration is mentioned once and then never discussed again. What role does it play? Is it important? Would tuning this better remove the need for the dist_max penalty?
- The notation $f_D(c_i)$ is unnecessarily complicated. These values correspond to states, no? So define them as $s_{c_i}$ or something similar to make the math and text more clear. As it is, equation (6) is unnecessarily cluttered.
- I assume that the argmax in the subgoal computation at the end of section 4 should be an argmax over c?
- There is a typo in the Figure 3 caption (PointmMaze). Further, the text in the subfigures is illegibly small.
- Break lines 11 and 13 of algorithm 2 into two lines each.


*****************
After author response:

I appreciate the updates you've made to the paper to better flesh it out, including the diagrams, pseudocode, and additional ablations. I've increased my score accordingly. Regarding generalization, I fully agree that it can be difficult to show. That said, when it's advertised in the title of the paper, I expect it to be clearly shown in the paper itself. It seems that the language has been greatly toned down in the updated version. However, without entirely re-reviewing the paper, I am unable to fully recommend acceptance.

Aside: I would have appreciated responses to my questions directly so that I don't need to dig through the rewritten paper to find the answers to my questions.

---

> ### Author Response · Authors · 2020-11-23
> **Thank you for your review!**
>
> Thank you for your detailed and helpful review. Your main critique of the paper seemed to be difficulty in following the algorithm pipeline. In the original draft, not enough care was taken to explain the entire pipeline and how the whole system fits together. We have put in a lot of work in our revisions to help fix this problem, and to clarify what exactly the key steps in the algorithm are, and how they are executed in practice. We have moved the step-by-step algorithm into the main paper. We have also replaced the architecture figure with two separate figures (Figures 1 and 2) that draw a comparison between this method and MBRL, and present the algorithm in a more linear fashion that is easier to follow end-to-end. Figure 1 explains the high-level intuition and steps behind the method, with Figure 2 supplying more implementation details. We took time to clarify the loss function, how the optimization over the latent space works, etc. We are committed to making sure the work is fully reproducible, and will include code with the final as well as hyper-parameter tables and discussion in the appendix. We feel like these improved figures, the code, and the algorithm box we have included, all improve the ability to understand the algorithm pipeline.
>
> You also mention concerns about ablation studies you would like to see. We ran two additional ablation studies that help to better understand the impact of the graph search and planning algorithms used in the paper. We feel these substantially help to clarify what parts of the algorithm are responsible for improved performance.
>
> Finally, we note that showing meaningful generalization is often a difficult part of RL algorithms. It can sometimes be difficult to evaluate the difficulty of generalizing from one task to another, because our intuition for this sort of thing can be deceiving. While generalizing to longer task lengths on ant maze may seem easy in principle, it is actually quite challenging and consequently existing algorithms perform quite poorly in this regard. A similar result holds for the robotics task of putting the block into a box. While there is some overlap with the train distribution, it is not substantial. In many ways, the learned behaviors on the training distribution tend to induce the agent to avoid the box, making test time generalization quite challenging since the test time task is entirely about interacting with both the block and the box. The primary focus of this paper is to enhance an RL agent’s ability to do temporally extended reasoning, and we hope that our updated draft helps clarify our main motivation.

---

### Official Review · AnonReviewer1 · 2020-11-04
**Useful and efficient approach, but more ablations, generalizations, and environments should be examined.**

**Rating:** 5
**Confidence:** 5

**Review:**

This paper approaches long horizon planning by learning a sparse graphical representation. The proposed algorithm, L3P, proceeds by learning a latent space which enforces a distance measure, where this distance is learned to mimic the number of steps between states via a goal conditioned Q-function. A clustering algorithm is then used to represent this latent space through only a small, efficient set of latent landmarks. These landmarks are then connected if nearby via the distance, the estimate of which is refined via soft value iterations over the graph. L3P is demonstrated on a number of environments to be both data efficient and high performing compared to baselines.

The paper is clear and well written. The topic of graphical representations of state spaces to plan long horizons over via local predictions holds significant promise, as does sparsifying this representation. The latent space construction and sparsification procedures are reasonable, particularly tying the distance to the policy Q-function as in SoRB. A few notes on clarity:
- The algorithm should be moved to the main body of the paper. (to reduce space, potentially remove a row of images in Fig. 4, stretch figure 1 to use the full width)
- A video would be helpful of the full process, including the soft value iterations.
- What algorithmic parameters (e.g., latent space dimensionality, network sizes, training data and number of states from which the latent space was constructed) were used?
- How are L_rec and L_latent traded off?
- The tasks, e.g., Fig 2 and 5, should be shown before the results in Fig. 3.
- More intuition on the soft value iteration would be helpful.

The results on the shown tasks show clear benefits of L3P, particularly on the tasks where HER has most difficulty (likely the more long range tasks) L3P outperforms substantially. However, the results are somewhat lacking (1) on ablation, (2) demonstrating new environment generalization, and (3) showing significantly complex tasks.
1) The authors show ablation for number of landmarks and d_max, but not for algorithmic changes like the soft value iteration. How much does this procedure help versus the use of latent marks to sparsify the space? It would also be interesting (though not fully necessary) to see comparison to Savinov 2018a (SPTM), which uses a learned distance predictor rather than the Q-function as in SoRB and L3P.
2) It is not clear how this method would generalize to new environments, e.g., Section 5.5 in SoRB. This to me is key (particularly with generalization in the title, which I believe currently only refers to task length) and may be a challenge for L3P due to finding landmarks from limited coverage of a new scene.
3) L3P should be shown on higher dimensional and longer horizon problems, as the maze is fairly short and simple , as is the box task (as can be seen by HER’s performance). These do not push the boundaries of the algorithm. Higher dimensional problems may particularly challenge the landmark learning (e.g., if the landmarks must be learned too in a higher dimensional latent space)


__________

After author response:
I appreciate the author’s response. Previous topics:
1) The author’s add ablations on the hard vs. soft min during the graph search, the additional results are informative, but not conclusive. Given that the overall performance is similar, the authors need to demonstrate the soft-min’s benefits for each experiment and over more training seeds.
2) For generalization, the author’s confirm that this method is unable to generalize to new environments, though clearly it has other benefits in terms of data efficiency and robustness of solutions. I believe these are still important benefits, though it would be useful to discuss how generalization may be achieved.
3) For harder experiments, the authors note that the baselines perform poorly there and thus these tasks were not considered. Though reasonable that the baselines are unable to perform in such cases, harder experiments would show the limit of the proposed algorithm. It would be useful to see for instance how well it scales with dimensionality, how quickly the success rate falls off.

New comments:
a) I believe the title change away from Generalization is an improvement, though the algorithm name "WORLD MODEL AS A GRAPH" seems to not capture the novel aspects of this work. This name I believe would be more readily applied to search on the replay buffer or semi-parametric topological memory.
b) R2's point that much of the robustness may be a factor of choosing states further from the wall is an interesting one. It would be interesting to examine exactly *why* the method is robust.
Overall, I believe the paper is interesting and proposes some novel ideas that have benefit, it requires more thorough analysis, and thus I am leaving my score unchanged.

---

> ### Author Response · Authors · 2020-11-23
> **Thank you for your review!**
>
> Thank you for your detailed and constructive feedback! We have updated the draft to address many of your concerns. As we understand it, your main worries are threefold. 1) Some key ablation results were missing. 2) The demonstrated generalization was not strong enough. 3) The considered tasks could have been harder.
>
> We agree that key ablation studies were missing. First of all, we have added ablation studies looking at the impact of soft min in graph search. The simplest way to explain this component is that we are using a variant of Floyd algorithm for graph search, except that we replace the hard min with a soft min (details see Appendix A.2). We comparse against hard min (Floyd) and see that indeed the change does make a difference. Secondly, we also did ablations for the effect of our online planning algorithm (Algorithm 1). We see that again the chosen planning strategy was important to the algorithm’s final overall success. We do agree that this discussion is important for understanding why exactly the algorithm is doing better than SORB and other graph search algorithms.
>
> For your point about generalization, we agree that our current algorithm will struggle to reach the kind of generalization demonstrated in Section 5.5 of SORB. There are two reasons for this. The first is that this setting in SORB considers using offline data on new environments, whereas our algorithm would likely require direct interaction. The second is that our current algorithm is not trained to be able to generalize to new environments - there are no meta learning components in the current algorithm. The main focus of this paper is to enhance an RL agent’s ability to do temporally extended reasoning. As for generalization to new environments given offline data, we think this will require combining the current method with meta learning approaches, and we leave it for future work.
>
> Additionally, regarding generalization of our algorithm, we note that showing enough generalization is typically a bane of RL algorithms. Indeed, many papers simply train and test in the same environment. This speaks to the difficulty in engineering an environment where sufficient generalization can be shown without simply making the environment too hard. In our maze environment, the agent never does navigate from one end of the maze to the other during training. On average, only 200 or so timesteps are required to complete the training tasks. Meanwhile, over 500 timesteps are required for the test time task. Meanwhile, for the robotic task of putting the block in the box, we note that the generalization is difficult because the setting is quite adversarial. Normally, hitting the box ruins the agent’s trajectory. So it has a large incentive to avoid that obstacle. Nevertheless, the agent is still able to generalize to test-time behaviors involving the box. The other comparable baselines did quite poorly on this task, taking an order of magnitude more data to achieve much worse test time performance.
>
> About evaluation on harder environments: we did consider harder multi-stage robotics tasks. However, we found that the baseline algorithms achieved near zero percent success in these harder environments. The chosen environments were selected because baseline algorithms were still able to achieve reasonable performance. While the ant maze does seem trivial, in a sparse reward setting it can be quite difficult -- as we can gauge by the poor performance of other methods. For the final, we are willing to include one of these more difficult environments if that would be persuasive.
>
> Finally, we have done our best to address all minor points you raised. The algorithm has been moved into the main paper. A discussion on algorithmic parameters has been added. The discussion on soft value iteration has been updated, making a more straightforward connection to Floyd and further explaining its role in the overall model. Section 4.3 of our method section, Section 5.4 of our experiment section, and the Appendix have been significantly improved in light of your comments.

---

### Decision · Program_Chairs · 2021-01-07
**Final Decision**

**Decision:**

Reject

**Comment:**

This paper proposes a model-based RL algorithm which, instead of simply fitting a parameterized transition model and uses rollout for planning, learns latent landmarks via distance-based clustering and conducts planning on the learned graph. Although some of these ideas themselves have appeared in literatures, the overall approach is very nice, novel and sophisticated. The experimental results appear strong and interesting. Most reviewers feel positive about the contributions of the paper, but there remain concerns that need to be addressed.

The proposed approach is highly nontrivial, and more ablation, generalization and environments need to be studied to fully justify what's going on. The authors agree to expand the paper and add the needed results, which would require substantial work thus reviewers recommend that the paper be submitted again to a future conference and receive another round of review. Showing the generalization is nontrivial, and it would be make the paper stronger if the authors put more thoughts into this issue, although it is not a must.

Minor: Another technical comment is that the approach seems heavily rely the choice of embedding distance. Learning the best embedding with meaningful embedding distance has been considered in other scenarios, see eg https://arxiv.org/abs/1906.00302. It would be interesting to try out and compare difference choices of the embedding distance.